

**Optimizing the precision of infrared measurements using the Eppley**
**Laboratory, Inc. model PIR pyrgeometer**
**Joseph J. Michalsky[1], John A. Augustine[1], Emiel Hall[1], and Benjamin R. Sheffer[1, 2]**
[1]Global Monitoring Laboratory, National Oceanic and Atmospheric Administration,
Boulder, Colorado 80305, USA
[2]Cooperative Institute for Research in Environmental Sciences, University of Colorado,
Boulder, Colorado 80309, USA
**Correspondence:** Joseph Michalsky (joseph.michalsky@noaa.gov)
**Abstract.** The Eppley Model PIR is widely used for thermal infrared wavelength (3.5-50 μm)
measurements of the downwelling and upwelling radiation from the atmosphere and surface,
respectively. The field of view of the instrument is $2\pi$ steradians with a receiver that has an
approximate cosine response. In this paper we examine four equations in the literature that have
been used to transfer calibration from standards to field units that are used for network
operations. After the introduction we discuss various equations used to convert the resistance of
the YSI 44031 thermistors used in PIRs for temperature measurements of the body, aka case, and
dome that are used in the derivation of incoming irradiance. We then use the four related, but
distinct, equations for the transfer of the calibration from standards to field instruments. A clear
choice for the preferred equation to use for calibration and transfer of calibration to field PIRs
emerges from this study.
**1. Introduction**

The Eppley model PIR pyrgeometer was developed to measure longwave thermal infrared (IR)
radiation emitted by the sky and surface. It originally came equipped with a battery-powered
circuit to compensate for the radiation emitted by the body, aka case, so the net signal from the
instrument was a measure of actual incoming infrared radiation. Users of this instrument that are
interested in maximum accuracy do not use the battery-powered circuit, but, instead, used
temperatures from two thermistors connected to the body and dome of the instrument along with
the thermopile output to calculate the incoming infrared irradiance signal.

Fig. 1 illustrates the most significant incoming and outgoing IR irradiances at the thermopile
surface. To derive $L$, the incoming IR radiation from the hemisphere outside the instrument,
radiative equilibrium of the instrument must be defined. To do that, the sum of the incoming
radiation transmitted by the dome (labeled 1), radiation emitted by the dome (labeled 2), and the
radiation emitted by the thermopile surface and reflected by the dome (labeled 3)  are set equal to
the radiation emitted by the thermopile surface (labeled 4). Considering these components
Albrecht and Cox (1977) formulated Eq. (1) for the externally received infrared radiation as





$$L = U_{thermopile}(c_1 + c_2 T_B^3) + \varepsilon_o \sigma T_B^4 - k\sigma(T_D^4 - T_B^4), \tag{1}$$


where $L$ is the external incoming infrared irradiance, $U_{thermopile}$ is the voltage measured across the
thermopile, $T_B$ and $T_D$ are the body and dome temperatures in degrees K, $\sigma$ is the Stefan-
Boltzmann constant, $\varepsilon_o$ is the emissivity of the detector, and $c_1$, $c_2$, and $k$ are constants to be
determined in calibration.

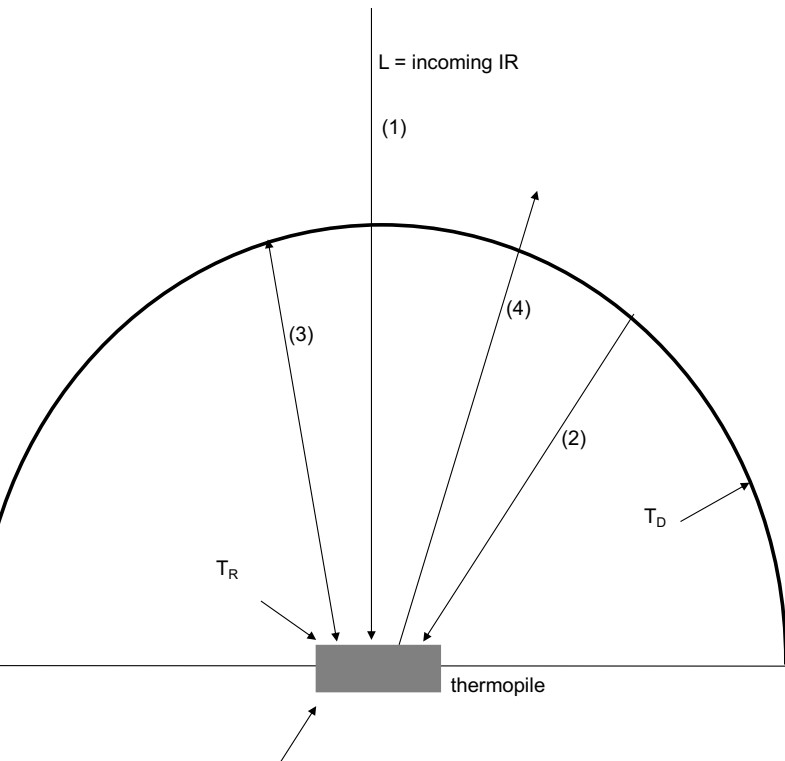

**Figure 1.** Schematic for the most significant incoming and outgoing infrared rays that are
considered in calculating the incoming infrared. The top of the dark rectangle is the receiving
surface surrounded by the dome that transmits infrared in the range 3-50 μm. $T_B$ and $T_D$ are the
measured body and dome temperatures in degrees K. $T_R$ is the estimated receiver temperature in
degrees K.
In practice Albrecht and Cox (1977) dropped the $c_2 T_B^3$ term as negligible relative to the $c_1$ term
and set the emissivity of the body of the instrument $\varepsilon_o$ to 1 yielding this commonly expressed
form of their equation

$$L = \frac{U_{thermopile}}{C} + \sigma T_B^4 - k\sigma(T_D^4 - T_B^4), \tag{2}$$






where $c_1$ has been replaced by $1/C$.

Philipona et al. (1995), however, used Eq. (1) in its entirety, but to compare symbolically to Eq.
(2) it is written

$$L = \frac{U_{thermopile}}{C}(1 + k_1\sigma T_B^3) + k_2\sigma T_B^4 - k_3\sigma(T_D^4 - T_B^4),\qquad(3)$$


where the $T_B^3$ term in Eq. (1) is retained, the emissivity of the body is $k_2$, and $k_3$ is the same as $k$
in Eq. (2*)*. All constants, C, $k_1$, $k_2$, and $k_3$, are determined in calibration.


Payne and Anderson (1999) used the functional form of Eq. (2), but substituted $T_R$ for the $T_B$,
where $T_R$ is the empirically calculated approximate temperature of the receiving surface rather
than the measured body temperature as illustrated in Fig. 1. Thus,

$$L = \frac{U_{thermopile}}{C} + \sigma T_R^4 - k\sigma(T_D^4 - T_R^4).\qquad(4)$$


Payne and Anderson (1999) estimated $T_R$ using Eq. 5

$$T_R = T_B + 0.694 \cdot U_{thermopile}\qquad(5)$$


where $U_{thermopile}$ is in millivolts, and the emissivity $\varepsilon_o$ is set to unity.

Reda et al. (2002) used a form similar to Eq. (4)

$$L = k_0 + \frac{U_{thermopile}}{C} + k_2\sigma T_R^4 - k_3\sigma(T_D^4 - T_R^4),\qquad(6)$$


where the instrument body emissivity $k_2$ is derived during calibration and a constant term $k_0$ is
introduced. $T_R$ is nearly the same as Eq. (5) with 0.704 replacing the constant 0.694. In this paper
we drop the constant term $k_0$.

The organization of this paper is as follows. Because accurate internal thermistor temperatures
are critical to pyrgeometer IR measurements, we first examine various versions of the Steinhart-
Hart equation that have been used to convert the YSI 44031 thermistor resistance to the
temperatures of the PIR body and dome. We then calibrate three test PIRs by transferring the
calibrations of our three standard PIRs that were in turn calibrated at the World Radiation Center
(WRC) in Davos, Switzerland, using the World Infrared Standard Group (WISG). Comparisons
are then made between the mean irradiance of the three standard PIRs and the computed
irradiance from the test PIRs using the four different forms of the original Albrecht and Cox
(1977) formula, i.e., Eq. (1) to calibrate each. Boxplots are used to demonstrate the level of
agreement between the standards and test PIRs for the various formulations. A clear conclusion
with regards to the preferred technique to use for calibrations and field measurements is
suggested in the summary.





**2. PIR Temperature Measurements**
The body and dome temperatures in the Eppley PIR pyrgeometer are measured using the YSI
44031 thermistor. Steinhart and Hart (1968) found that a cubic fit of inverse temperature to the
log of measured resistance matched many thermistor data points over a wide temperature range.
Their equation is

$$\frac{1}{T} = a + b \cdot \ln(R) + 0 \cdot \ln(R)^2 + d \cdot \ln(R)^3, \tag{7}$$

where $T$ is in degrees K and $R$ is in ohms or kiloohms. Note that there is no squared term in the
standard Steinhart-Hart equation. Coefficients $a$, $b$, and $d$ differ depending on whether ohms or
kiloohms are used and depending on the temperature range over which the fit is made.
Fig. 2 is a plot of four independently-derived fits to the YSI 44031 thermistor data. The y-axis is
the temperature estimate based on the fit minus the tabulated thermistor data to which the fit is
made. The least-squares fit to Eq. (7) (no quadratic term) is the red line if ohms are used. If a full
cubic, including a quadratic term, is used to fit the tabulated data in kiloohms, then similar, but
not identical, agreement is obtained (blue-white line). Interestingly, if the fit with the full cubic is
made to ohms, rather than kiloohms, the agreement is identical (again, note the blue-white line).
This is not the case if the quadratic term is not included in the fitting to ohms versus kiloohms as
will be discussed in the appendix. Two fits found in the literature (McArthur, 2005; Gröbner,
2025) that are fit to the resistance in ohms are plotted in Fig. 2. The Physikalisch-
Meteorologisches Observatorium Davos (PMOD) fit (Gröbner, 2025) was over a -30 to +40 °C
range, but does well over the entire range.  If the Baseline Surface Radiation Network (BSRN;
McArthur, 2005) coefficient $a$ is modified slightly from the published 0.0010295 to 0.0010293,
as shown in the legend, then an improved fit (upper gray curve) is obtained that agrees well with
the others in Fig. 2. Although the differences among the various fits in Fig. 2 are small, the full
cubic is used to compute PIR temperatures in the data analyzed here.

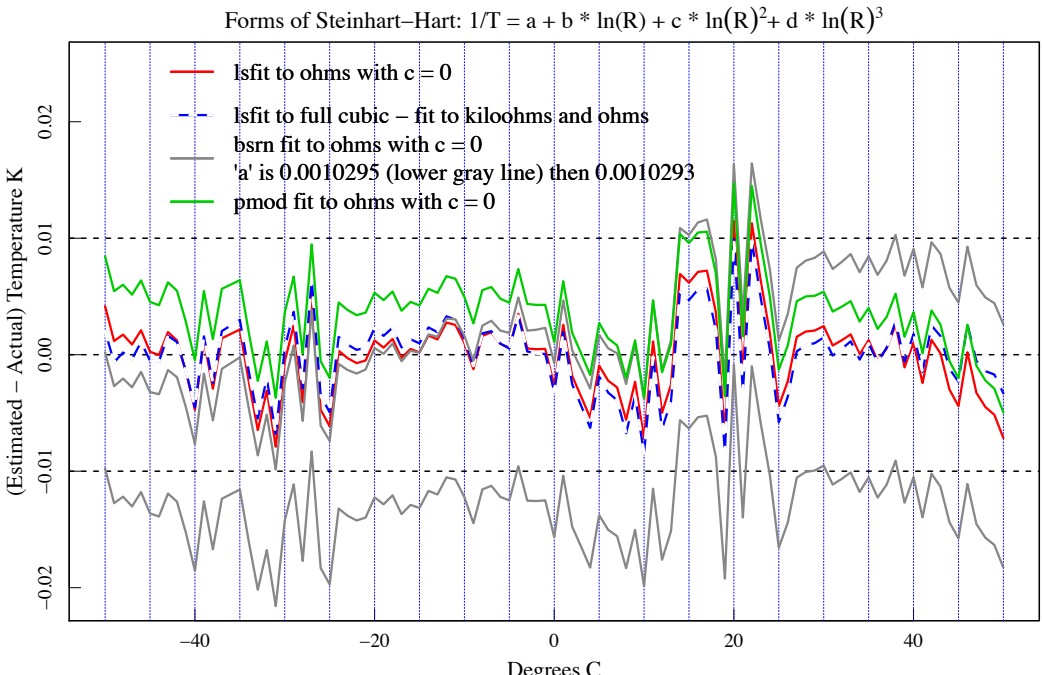


**Figure 2.** Four independent fits using forms of Eq. (7) to the YSI 44031 tabulated data after
subtraction of the tabulated data over the range -50 to 50 °C in 1 °C increments. Similar
agreement among all four ensues if the small change to the published BSRN constant *a* is made.

## 3. Four Methods of PIR Calibration Transfer

In this section, we apply Eq. (2), (3), (4), and (6) to examine how well each performs in
transferring calibrations from our three "standard" PIRs, which were calibrated against the world
reference at PMOD, to field PIRs. Our three PIRs were calibrated at the World Radiation Center
(WRC), aka PMOD, in Davos, Switzerland in 2018, 2022, and 2024. ([https://www.pmodwrc.](https://www.pmodwrc.ch/en/world-radiation-center-2/irs/wisg/)
[ch/en/world-radiation-center-2/irs/wisg/](https://www.pmodwrc.ch/en/world-radiation-center-2/irs/wisg/)). Each PIR was returned with two sets of coefficients,
one set for Eq. (2) and one set for Eq. (3).

To transfer the standards' calibration to field radiometers, the standards and test PIRs are
arranged side-by-side for a week or more on an outdoor horizontal observing platform, with no
significant obstructions surrounding the platform. Fig. 3 is an example of one such calibration
period. On the left are the three standards' outputs with the WRC's Philipona et al. (1995)
coefficients applied. On the right are the same standards with the WRC's Albrecht and Cox
(1977) coefficients applied. Agreement among the three on the left is very good because the last
PIR readings (green) overplot the first two (black and red). Agreement on the right is nearly as
good but with some underestimation by PIR 32909F3 (red dots).

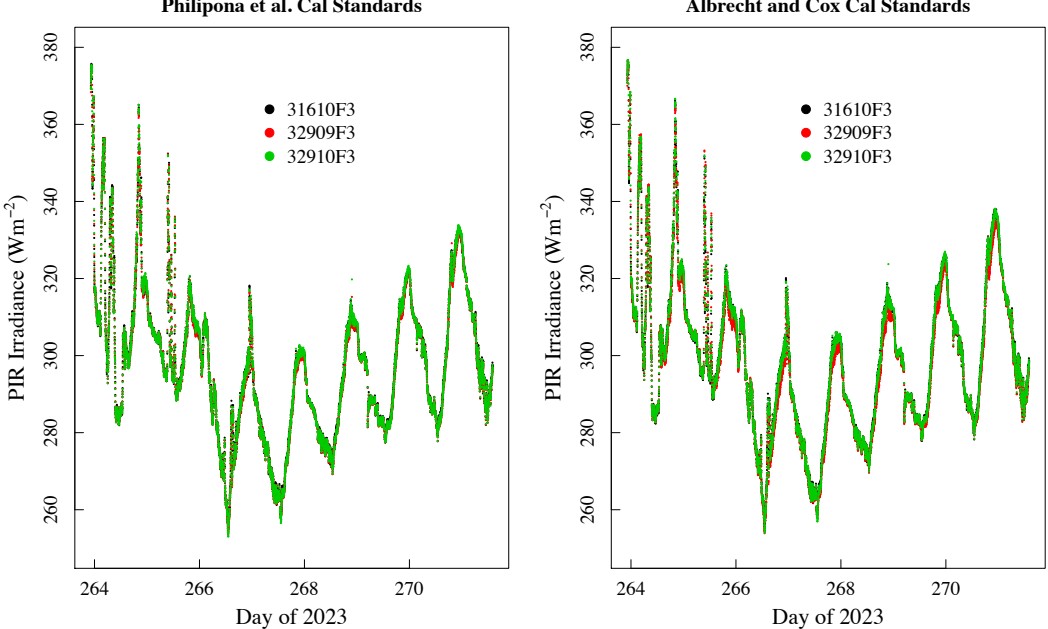

**Figure 3.** Calculated IR irradiance from our three standard PIRs (serial numbers in the legend)
using Philipona et al. coefficients provided by WRC are overplotted on the left and the Albrecht
and Cox coefficients are used on the right. This demonstrates the agreement among the standards
using the two methods. The number of days used is typical for our calibration runs.
Before comparing results from Eq. (2), (3), (4), and (6), we first compare results from only
Albrecht and Cox (Eq. 2) and Philipona et al. (Eq. 3) for which the WRC provided both sets of
coefficients. In this test, the mean IR irradiance of the three standards is compared to computed
IR from a test PIR that was calibrated using these standards. The least-squares fitting technique
to determine the calibration coefficients for the test instrument uses a robust function in the R
language (MASS::rlm) that de-weights outliers to reduce the effects of noisy, for example, rain-
contaminated, as well as other outlier data.
In Figure 4, boxplots are used to compare the performance of the two calibration methods
applied to the standards calibrated at the WRC. The "box" in these plots contain 50% of the data,
and the lines extending from the top and bottom of the box, or "whiskers," include about 95% for
normally distributed data. In the top-left panel of Fig. 4 the three standards use the WRC-
provided Albrecht and Cox (1977) coefficients, and the average of the three standards is
compared to coincident test PIR (SN 23215F3) data, also calculated using Albrecht and Cox
(1977). The boxplot summarizes those differences over the entire calibration period for PIR
23215F3. The boxplot on the top right summarizes differences following the same procedure, but
using Philipona et al. (1995) coefficients for the standards (WRC-provided) and for the test PIR.
Comparing the top panels of Fig. 4, the one on the right, where Philipona coefficients are used
exclusively, has a smaller box, shorter whiskers, and a median nearer to zero compared to the
panel on the left where Albrecht and Cox was used exclusively.



The bottom panels of Fig. 4 show the same comparison for a different test PIR (SN 38805F3).
The same comments apply, with the Philipona et al. (1995) calibrated data (bottom right) giving
smaller spread in the box and whiskers, and the median nearer zero, while there is more spread in
the bottom-left panel where Albrecht and Cox (1977) is used. Differences in the lower panels of
Fig. 4 are generally greater than those in the top panels. The calibration data for these two test
instruments were collected concurrently, which suggests that the disparity arises from inherent
characteristics of the instruments themselves. We studied a total of six instruments from two
distinct calibration periods in this way and found that in every case using the Philipona et al.
(1995) form (Eq. 3) gave better results than the formulation of Albrecht and Cox (1977) (Eq. (2).

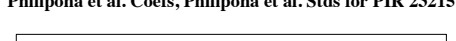

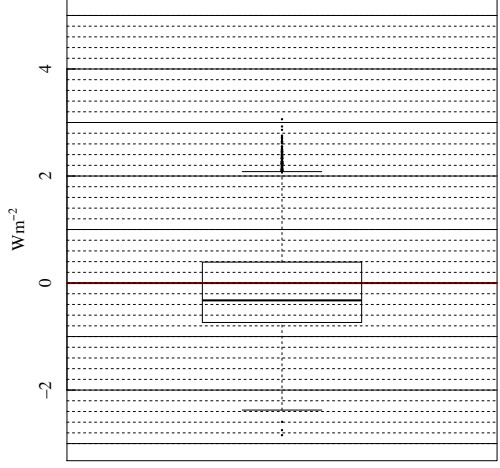
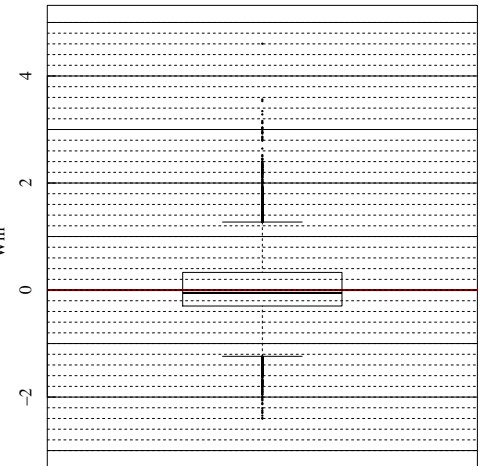






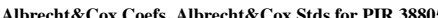

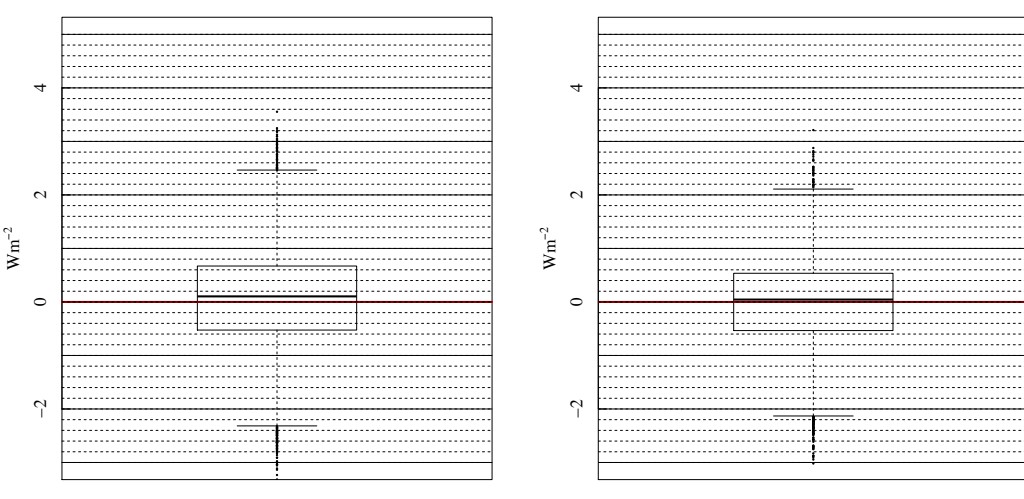

**Figure 4.** (top) Boxplots of the differences between applying Albrecht and Cox (1977)
calibrations and applying Philipona et al. (1995) calibrations for PIR 23215. Note the differences
in box widths, whisker lengths, and median values. (bottom) Boxplots for a different PIR
(38805) that was calibrated at the same time as the one in Fig. 4 (top).
Next, we compare results from all four equations (2), (3), (4), and (6) for the same two PIRs as in
Fig. 4. Since Fig. 4 suggests that the Philipona et al. (1995) Eq. (3) produces better results than
Albrecht and Cox (1977) Eq. (2), we will use Philipona et al. (1995) coefficients provided by
WRC to compute IR irradiance for the standards and average these as "truth" for all of the
comparisons. For both test PIRs in Fig. 5 (top and bottom) the last boxplot on the right
(Philipona et al., 1995) gave the best results followed by the adjacent boxplot (Reda).




Left to Right, Albrecht Coefs, Payne Coefs, Reda Coefs, Philipona Coefs Using Philipona Stds for PIR 23215 2023 Cal

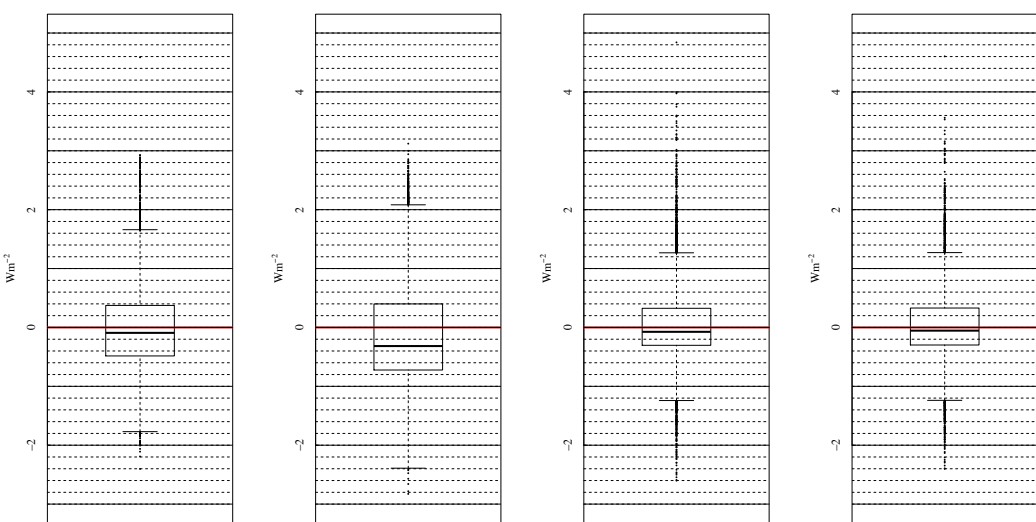

Albrecht = Albrecht and Cox (1977); Payne = Payne and Anderson (1999); Reda = Reda et al. (2002); Philipona = Philipona et al. (1995)


Left to Right, Albrecht Coefs, Payne Coefs, Reda Coefs, Philipona Coefs Using Philipona Stds for PIR 38805 2023 Cal

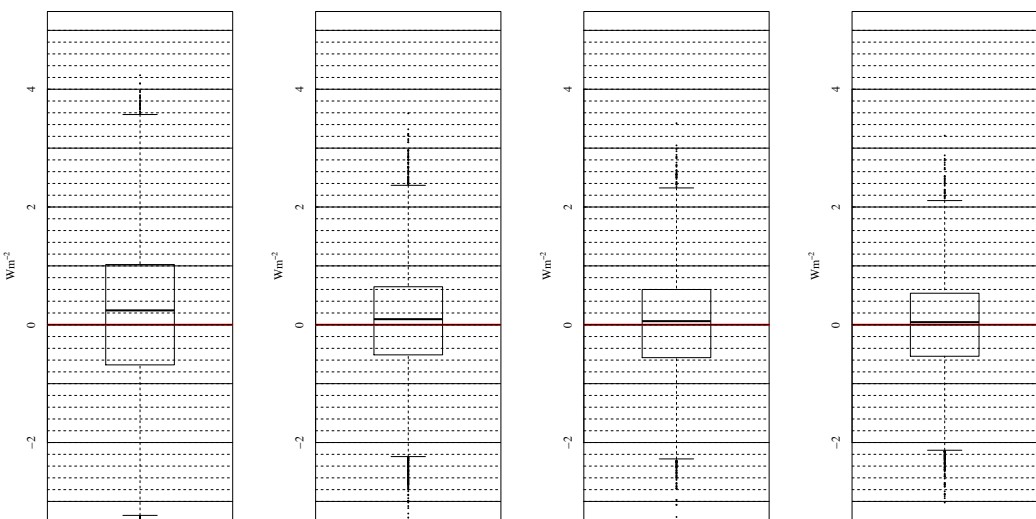

Albrecht = Albrecht and Cox (1977); Payne = Payne and Anderson (1999); Reda = Reda et al. (2002); Philipona = Philipona et al. (1995)

**Figure 5.** Boxplots of differences using the WRC's Philipona et al. (1995) coefficients for the
standards and calibrated PIRs to this standard using the four equations to calculate incoming
infrared. Top is for PIR 23215 and bottom is PIR 38805 as in Fig. 4. Compare box widths,
whisker lengths, and medians.
Note that the leftmost boxplots of Fig. 5 are similar to the leftmost boxplot in Fig. 4, with better
results for the top (PIR 23215), but worse results for the bottom (PIR 38805). In these two cases





the standard used was calibrated with Philipona et al. (1995) coefficients rather than Albrecht
and Cox (1977) as in Fig. 4. Similar results were obtained for the other four PIRs with the best
results always obtained with the Philipona et al. (1995) formulation. In only one case out of the
six the Payne and Anderson (1999) formula performed slightly better than the Reda et al. (2002)
formula (not shown).
**4. Precision of the PIR Standards**
Four formulae that are in the literature for calculating incoming infrared with an Eppley model
PIR pyrgeometer were tested to assess the precision of transferring calibrations from standards to
field deployed PIRs. Calibrations of the same three standards were made at the WRC in 2018,
2022, and 2024. For each of these independent calibration events, coefficients for the Albrecht
and Cox (1977) and Philipona et al. (1995) forms of the PIR processing equation for calculating
incoming infrared were provided by the WRC. Here we look at the repeatability of those
calibration events.
Our calibration seasons typically run from late Spring to early Fall. Therefore, our three PIR
standards experience roughly six months of exposure to the weather each year. In Fig. 6
differences from applying three sets of WRC Philipona calibration coefficients (from 2018,
2022, and 2024) to the same dataset (that used for Fig. 3) are summarized. For example,
calibrations from 2018 and 2022 were applied to the same dataset and differences in irradiance
for each minute were tallied and summarized in boxplots. Differences for all permutations are
mostly within 1 Wm$^{-2}$ and suggest that errors from applying one of the WRC calibrations from
any of the three calibration years to any year would be less than the uncertainty of the WRC
calibrations themselves (~ 4 Wm$^{-2}$; https://www.pmodwrc.ch/en/?s=wisg). This suggests that the
Eppley PIR is very stable and should be suitable for monitoring long-term changes in the thermal
infrared.





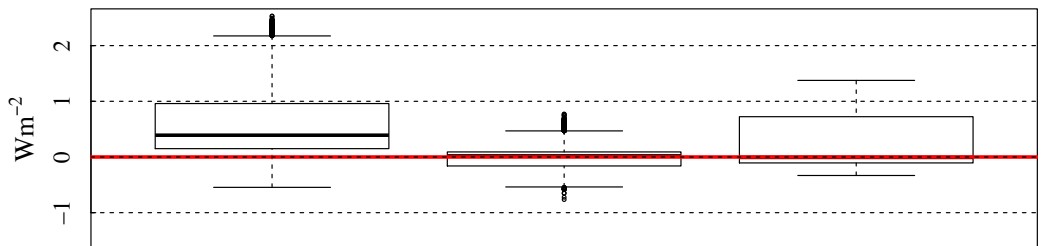

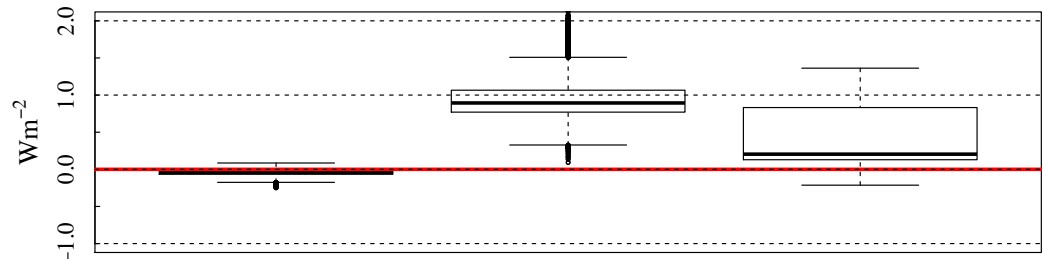

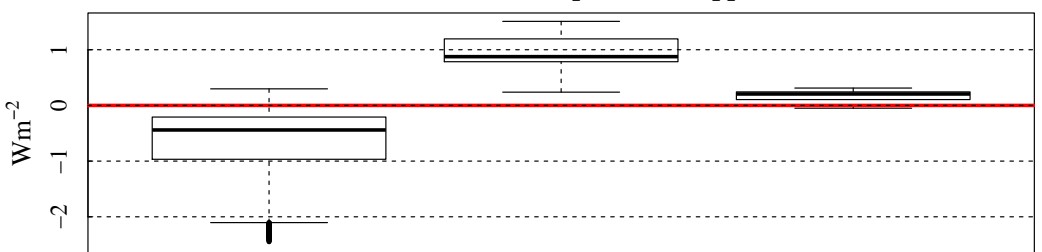

Three Standards (31610, 32909, 32910) w/ WRC Cals from Three Different Years

**Figure 6.** Comparisons of three sets of Philipona et al. (1995) calibration coefficients provided by the WRC in 2018, 2022, and 2024 applied to the same data set as in Fig. 3 for the three PIRs used as standards with serial numbers in the subtitle. The medians are all within 1 Wm$^{-2}$ and most are within 0.5 Wm$^{-2}$.

In section 3 the average output of the three standards is used to derive new calibration coefficients for each test PIR. Using those new calibrations, the test instrument measurements are compared to the standards' average over the entire calibration period. For the left panels in Fig. 7 we use Philipona et al. (1995) coefficients for the standards to calibrate the three test PIRs (serial numbers shown at the top of each subplot). We apply those new calibrations and subtract




the results from the standards' average for each minute and summarize the distribution of
differences in boxplots. Therefore, the leftmost panels of Fig. 7 replicates the rightmost panels of
Figs. 4 and 5. This is not an independent test of the reliability of the calibration because the same
dataset is used for calibration and verification.
To test new calibrations with an independent dataset, the time series in Figure 3 is divided in
half. The middle panels of Fig. 7 use the first half of the data in Fig. 3 to derive a calibration and
the second half of the data to validate the new calibration against the standards' average. Then,
we reverse this process using the second half of the Fig. 3 data for calibrating and the first half to
validate. If we examine the time series in Fig. 3, it is apparent that the first half of the data stream
is noisier than the second half. Using the first half of the data to calibrate and applying to the
second half and vice versa is likely responsible for the offsets in the medians, but the offsets are
less than one Wm$^{-2}$. Note that when the less noisy data of the second half are used to validate
(middle boxplots) the differences have a smaller spread. When the noisier first half data
(rightmost boxplots) are used to validate, the differences have a larger spread. Examining the
top, middle, and bottom plots, there are differences inherent in the instruments themselves since
boxplots are not replicated from PIR to PIR. Attribution to the instruments themselves is
warranted because the standards and test data used for Fig. 7 were collected simultaneously.

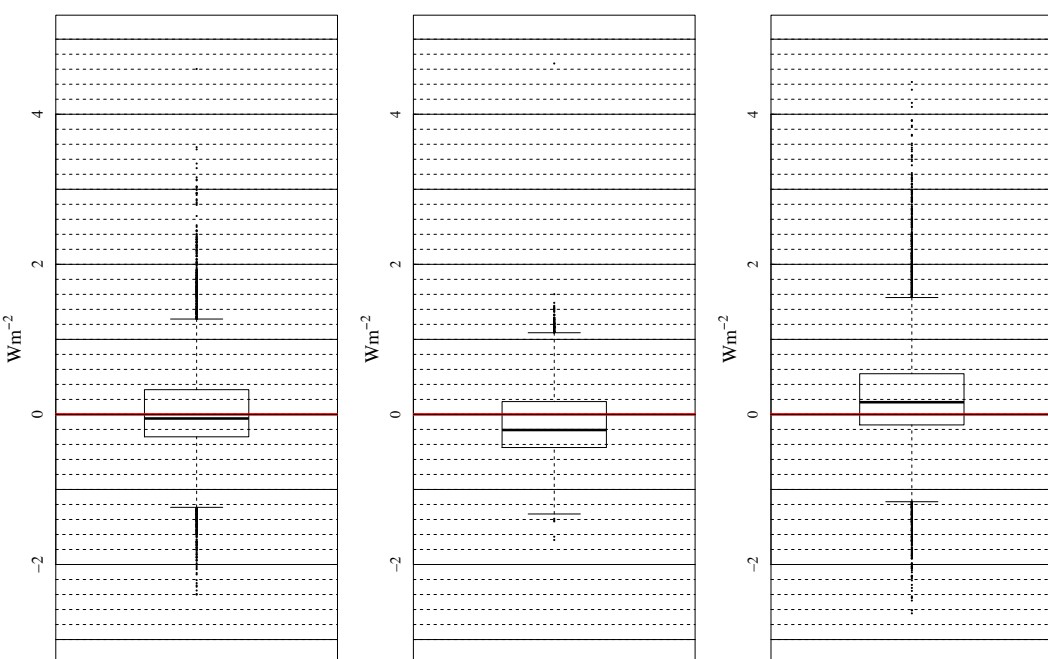

Left to Right: Full Cal Period; 1st Half Used for Cal, Applied to 2nd Half; and Vice Versa for PIR 23215




Left to Right: Full Cal Period; 1st Half Used for Cal, Applied to 2nd Half; and Vice Versa for PIR 28139

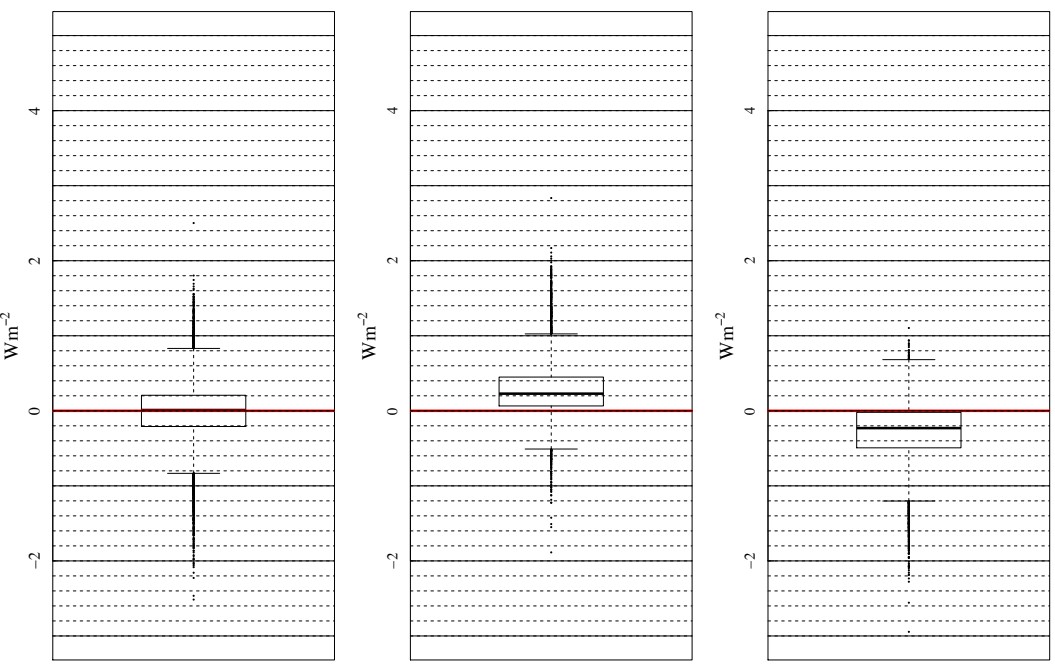


Left to Right: Full Cal Period; 1st Half Used for Cal, Applied to 2nd Half; and Vice Versa for PIR 38805

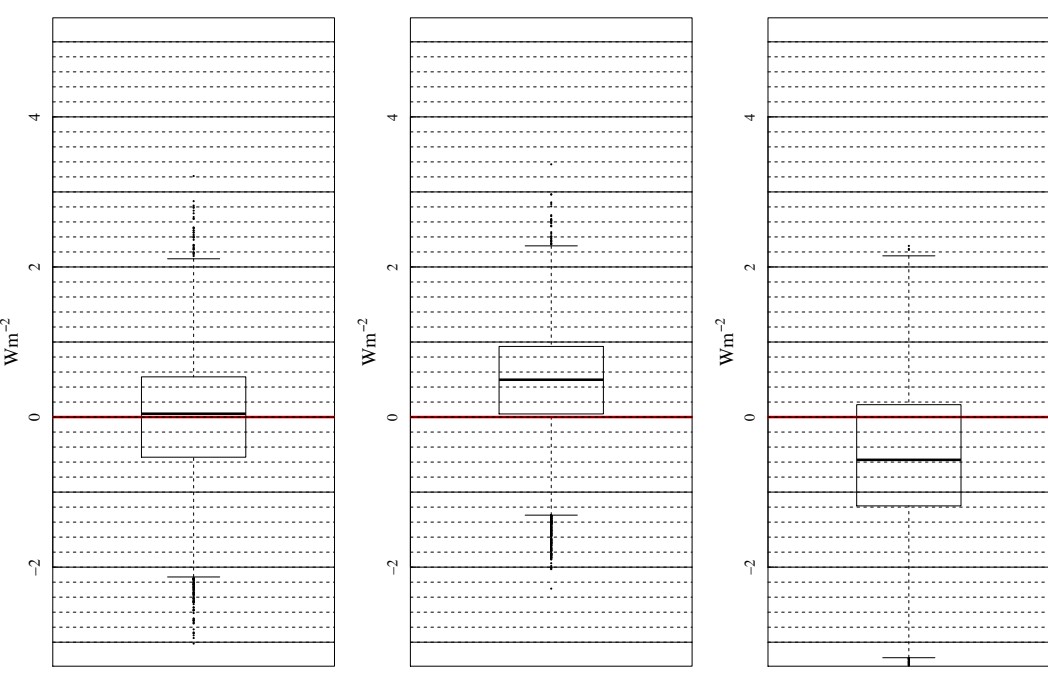




**Figure 7.** The leftmost panel uses the entire period in Fig. 3 to calibrate the named PIR and then compares the calibrated PIR data to the standards averaged. The middle panel uses the first half of the period to calibrate and then compares the second half using these calibrations. The rightmost panel reverses this using the second half of the period for calibration and the first half for comparisons.

**5. Summary and Conclusions**

In this paper we investigate four formulations for converting raw voltage and body and dome temperature measurements of an Eppley pyrgeometer, model PIR, to thermal infrared irradiance. These methods are described in Albrecht and Cox (1977), Philipona et al. (1995), Reda et al. (2002) and Payne and Anderson (1999). All are slight variations of the original formulation of Albrecht and Cox (1977). Because the temperature measurements are critical to the calculations, we also investigate various fits that have been applied to the Steinhart-Hart (1968) equation that converts thermistor-measured resistance to temperature.

Regarding the computation of thermistor temperatures, we found that fitting the manufacturer-supplied table of resistance and temperature (1°C interval) to the range -50° to 50°C provides the least variability as opposed to fits to shorter temperature ranges. However, differences of the fit to the provided data are < 0.01°C, regardless of the range used. Based on this result, we conclude that differences in thermistor temperature calculations from fits based on various temperature ranges do not have a significant impact on PIR measurements.

The three standard PIRs that we use to transfer calibrations from the world standard to field PIRs are sent biennially to be calibrated against the world infrared standard group (WISG) at the World Radiation Center in Davos, Switzerland. They are returned with calibration coefficients for the Albrecht and Cox (1977) and Philipona (1995) methods, although the Albrecht and Cox coefficients provided are for the shortened form of their equation (Eq. 2). Comparing the application of the two methods to the standard PIRs revealed that the Philipona (1995) method is more accurate and less noisy than the Albrecht and Cox formulation, however, the differences are small. Comparisons were also made among three distinct WRC calibration results for the standard PIRs in 2018, 2022, and 2024. They showed that the three standard PIRs are stable, with the calibration coefficients changing minimally between WRC calibrations, and differences in irradiance calculations among applications of the separate biennial calibrations are within one Wm$^{-2}$ of each other.

Lastly, application of the four methods for converting PIR raw measurements to irradiance was analyzed using six test instruments. The major conclusion is that use of the Philipona et al. (1995) form, i.e., Eq. (3), consistently does the best in transferring the mean calibration of the standards to field-deployed PIRs. Note that Reda et al. (2002) and Payne and Anderson (1999) coefficients are not available for the calibration standards, which may have led to some of the differences in Fig. 5. Of the six calibration comparisons, like those in Fig. 5, Reda et al. (2002) calibration results were very close to Philipona et al. (1995) results on two occasions, while in one comparison Payne and Anderson (1999) results were very close to those of Philipona et al. (1995). However, this agreement was not consistent for the other four PIRs.





In this paper the World Infrared Standard Group (WISG), which is used for calibration at the
WRC, is the current standard for broadband infrared measurements. Recent studies, which are
summarized in Gröbner et al. (2024), suggest that the current WISG may be low by as much as 4
Wm⁻² if the water vapor column exceeds 1 cm, but the difference is smaller if the atmosphere is
dryer approaching no difference for vanishing water vapor (see Fig. 2 in Gröbner et al., 2024).
Nevertheless, a new standard for broadband infrared radiation is not expected to be established
until the next WMO congress in 2027 at the very earliest (Laurent Vuilleumier, private
communication).

**Appendix A1**


Fitting the manufacturer-supplied temperature (at 1 °C intervals) and resistance data, separately
in ohms and in kiloohms, led to an unexpected outcome. First, if a full cubic (i.e., non-zero
coefficient for the squared term) least-squares fit of the Steinhart-Hart equation with YSI 44031
data in kiloohms is compared to a least-squares fit using ohms, identical fits are obtained (blue
and white dashed line in Fig. 2). If the quadratic term is set to zero and the fits are made to ohms
and then kiloohms, we see a significant difference as shown in Fig. A1. The source of the
difference shown in Fig. A illustrates numerical differences when Eq. 7 (no squared term) is
analyzed using ohms versus kiloohms. Note the relatively large error when kiloohms are used.
Here we explain how that difference comes about.

First, it must be noted that the lack of significant digits when using kiloohms is not an issue
because for the fits here, kiloohms are computed simply by dividing the resistance value in ohms
by 1000, keeping significant digits in the decimal places.

The requirement of a quadratic term for expressing the Steinhart-Hart equation in kiloohms can
be demonstrated by substituting for $R$ in Eq. (7) $1000R_k$, where $R_k$ is in units of kiloohms as
shown in Eq. (8).

$$\frac{1}{T} = a + b \cdot ln(1000R_k) + d \cdot ln(1000R_k)^3 \qquad (A1)$$


Applying logarithm rules to Eq. (8) results in Eq. (9).

$$\frac{1}{T} = a + b\big(ln(1000) + ln(R_k)\big) + d\big(ln(1000) + ln(R_k)\big)^3 \qquad (A2)$$


Expanding and regrouping terms in Eq. (9) then gives Eq. (10) through Eq. (14).

$$\frac{1}{T} = a_k + b_k \cdot ln(R_k) + c_k \cdot ln(R_k)^2 + d_k \cdot ln(R_k)^3 \qquad (A3)$$


$$a_k = a + b \cdot ln(1000) + d \cdot ln(1000)^3 \qquad (A4)$$


$$b_k = b + 3d \cdot ln(1000)^2 \qquad (A5)$$




$$c_k = 3d \cdot ln(1000) \tag{A6}$$
$$d_k = d \tag{A7}$$
Thus, when data are in kiloohms an equation of the form of Eq. (A3) (i.e. full cubic) is required
to match the results of Eq. (7) when data are in units of ohms. Thus, changing units of $R$ in Eq.
(7) results in a full cubic equation. This implies that a full cubic equation can be more robust
than Eq. (7) when fitting data where units other than ohms are used for $R$. It also demonstrates
that it is possible to change units for $R$ in Eq. (7) analytically using the substitution process
shown above rather than refitting if desired.

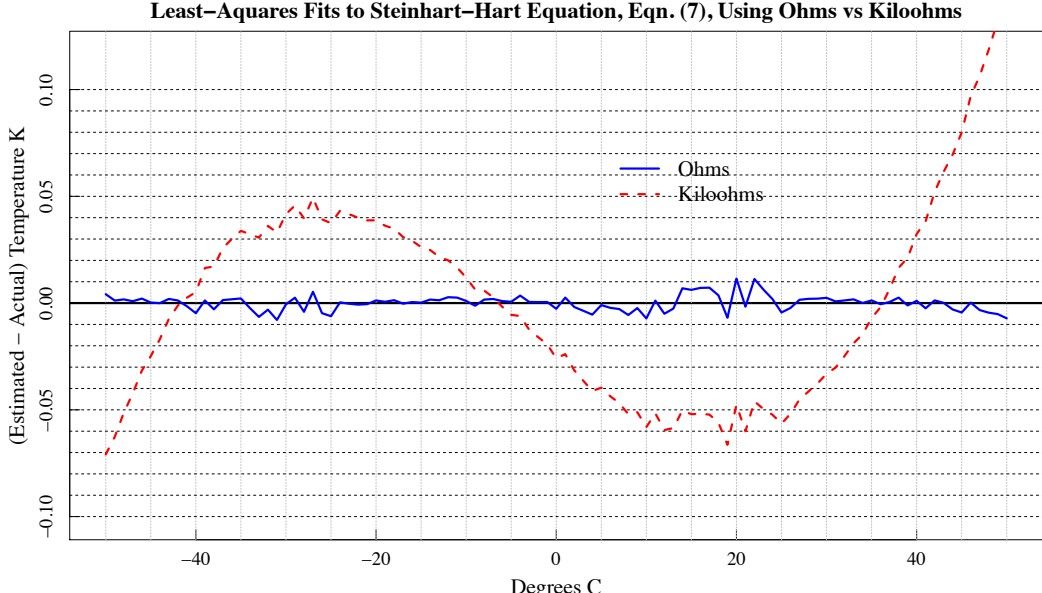

**Figure A1.** Steinhart-Hart equation fit to ohms (blue solid line) versus kiloohms (red dashed
line).

*Code availability.* Codes used to generate the results in this paper were original functions written in the
programming language R and are available by contacting joseph.michalsky@noaa.gov.
*Data availability.* Data can be made available by contacting joseph.michalsky@noaa.gov.



*Author contributions.* JJM did most of the analyses, drafted the paper, and produced the figures. JAA
provided the World Radiation Center calibrations and much useful discussion of the results. EH provide
the experimental data from the calibration table used for these analyses. BRS did the analysis for and
wrote the Appendix. All authors read and offered corrections to parts of the manuscript.

*Competing interests.* The contact author has declared that none of the authors has any competing interests.

*Acknowledgments.* The authors would like to thank Kathy Lantz for a careful reading of this paper.

*Financial support.* This research has been supported by the Global Monitoring Laboratory of the National
Oceanic and Atmospheric Administration.

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
