# Peer review of "Optimizing the precision of infrared measurements using the Eppley Laboratory, Inc. model PIR pyrgeometer"

_EGUsphere, 2025_

## Author Comment (AC1)

"Optimizing the precision of infrared measurements using the Eppley Laboratory, Inc. model PIR pyrgeometer" by Michalsky et al. evaluates existing approaches for transferring calibrations from PIR standards to sensors used for field operations. The authors conduct a careful and transparent analysis that provides useful results for long term networks with a historical investment in the PIR sensor. The study is a good fit for AMT. I have some comments that should be addressed in a revision before publication.

General comments:

(1) What equation is used to produce calibrated fluxes in the WISG sensors? If the WISG also uses eq (3), then wouldn't the results shown here be interpreted firstly as an indication of consistency, but not necessarily an indication of accuracy?

Yes, the WISG sensors use Eq. (3) coefficients, which is, in fact, the original Albrecht and Cox (1977) formulation, i.e., Eq. (1). This paper is about precision transfer of calibration to other pyrgeometers, and which of four forms of Eq. (1) best does this. We have removed the occurrences of the word 'accuracy' in regards to irradiance from our text to clarify that this is about precision.

(2) Figure 7: Some additional analysis here is warranted. The most obvious candidate for explaining the difference between the first and second half of the cal period is the cloud fraction (the repeatability of pyrgeometer measurements is much worse under clear skies than stratiform clouds; e.g., https://doi.org/10.5194/amt-14-1205-2021), but perhaps mean temperature or precipitable water vapor could also explain it. Figures 6 and 7 could mean that the small differences reported earlier (from any equation) are overly optimistic compensation between opposing errors given fortuitous proportions of conditions within the cal period.

We agree with the referee that the small differences depicted in Figs. 6 and 7 are likely caused by outliers associated with changing meteorological conditions that were not excluded in either of the two halves of the data set. Our objective approach to exclude points was to use a robust fitting technique that reduces the influence of outliers without having to undertake a detailed, subjective approach for which data to exclude. In the end the differences are fairly small. The referee makes the point that in his study in the Arctic that the most stable conditions were for uniform stratus clouds. In fact, the WMO broadband infrared calibration transfer is performed under clear and stable conditions (Groebner--and--Wacker--2015--WMO120).

(3) It would be good to apply t-tests to determine which means are different from one another, or from zero, where appropriate. The analyzed differences are small enough that they may not be significant.

If one assumes that there are no significant differences in the calculation of infrared irradiances using the Philipona et al. (1995) formula versus each of the other three methods discussed in this

paper, this assumption is rejected with 95% confidence in 15 of the 18 cases studied (six calibrated PIRs and three formulae). The three cases where the null hypothesis cannot be rejected with 95% confidence are for three of the six PIRs using the Reda et al. (2002) formula. Changes have been made to the text.

(4) L320-322: Regarding conclusions, what about the fact that the differences amongst transfer equations is so much smaller than either the WISG uncertainty or (speculatively, see 2 above) the uncertainty caused by the sampling of conditions during outdoor calibrations?

The WISG uncertainty is 2.6 Wm$^{-2}$ (Groebner--and--Wacker--2015--WMO120). The median differences in all formulae used to transfer calibration from the WISG are considerably smaller than this. Further, Fig. 7, which shows differences caused by the failure to sufficiently de-weight all outliers are also much smaller than the WISG uncertainty. Undoubtedly, in non-stable conditions the uncertainties for field measurements will be larger. However, our goal in this paper is to minimize the uncertainty where we can, specifically, in calibration transfers.

Specific comments:

L14: For clarity, "broadband thermal IR…"

Change made to clearer terminology

L40: Maybe clarify that the dome is designed to partly transmit only in the range of 3.5-50 um.

A sentence was added to clarify the dome transmission issues.

Figure 1: A few suggestions to improve the communication in this figure: (a) Label "dome" in the picture as you have done with the thermopile so that it is not interpreted as schematic of example paths in the sky (as I did at first). (b) In the caption after the word "rays" clarify that these are the numbered vectors in the picture. (c) Tb is not actually at the base of the thermopile, but is potted in the bronze casing nearby, so it would be helpful to depict the upper part of the case to show that Tb and Tr are not measuring the same thing. (d) Label Td and Tb as being thermistor measurements to distinguish from Tr, which is estimated (see also my comment at L92, which could also refer back to this figure).

Fig. 1 has been changed according to referee's suggestions.

L88: Since it is not clear from this text what Reda et al.'s justification was for including k0, it is also not clear what the present study's justification is for dropping it.

Likely, the Reda et al. (2002) equation is used at the National Renewable Energy Laboratory, however, the ARM program, for which NREL calibrates PIRs does not use the constant term in their calibration transfer to field PIRs. That is why it is not used here.

L90: I think this paragraph would benefit from a leading statement expressing the problem this paper is solving. That statement might be supported by another that explains the reason prior studies modified the original Albrecht and Cox approach. As is, the text presumes too much insider knowledge on the historical context and current gap in understanding.

Simple introductory sentence added.

L92: The fact that YSI44031s are used to measure the temperature, and which temperatures are measured this way, should be included in Figure 1.

Added to the Fig. 1 caption.

L112: Eq. 7 is odd. Can you write "c" instead of 0 in the equation to be more consistent with the Section 2 analysis/figure and then clarify in the text that in the classical form, S-H set c = 0?

Done.

L119: I'm confused about the use of the quadratic term. It looks like c = 0 for all lines in Figure 2. Where in the figure is the full cubic found? If it is the dashed blue line, it seems to be defined differently, as there is a minus sign in both the legend and the y-axis (Is the dashed blue line actually comparable to the other lines?) Also, what is c when it isn't 0, and when it is not 0, are a, b, and d the same or do they also change?

Fig. 2 has been changed to clarify these points with added text in the caption.

L120: "Interestingly…" I don't understand this statement. It seems like it would be much more surprising that changing the units yields a different result. The paper is not very long. Perhaps the appendix can be returned to the main text.

Text was changed to clarify this point.

Figure 2: An error of 0.01 C in the thermistor will produce an error < 0.05 Wm2 at 0 C, which is negligible compared to other uncertainties (similar, in fact, to the error produced by the conventional, though incorrect, assumption that sigma is 5.6700e-8). Isn't it true that the most relevant problem attributable to the YSI44031 is not the calibration method, but instead either the representativeness of its placement in the sensor in the case or the variance amongst individual thermistors in conforming to the calibration coefficients? So, I'm left not being entirely sure what the purpose of this exercise is. Is the take-away message that the YSI calibration isn't the problem with the flux calibration? If so, make that clear. [Returning to this point after reading the conclusion, I appreciate the point you made at L299-304, though it might be worth commenting on the other issues with the thermistor in the conclusion. At very least, I suggest making the purpose of the thermistor section clearer in Section 2.]

The point about the uncertainty in the temperature measurements using a 0.2 K thermistor is added to the text. I know nothing about the uncertainty caused by thermistor replacement within the brass body.

L164: When you say "using these standards", do you mean that the average of the standards was used for the calibration?

Fixed.

L247, 266: I think Figures 6 and 7, which show larger differences than Figures 3 and 4, suggest that the conditions under which outdoor calibrations (clarify somewhere that these are indeed outdoor?) are carried out are responsible for larger calibration uncertainty than the choice of equation. Yet, I think the community has historically been more focused on methodology. Maybe a recommendation to be made there?

Paragraph added to summary section suggesting larger errors should be expected for general conditions.

L313: "…are small." Specifically, the differences are an order of magnitude smaller in the transfer of relative calibrations than the reported uncertainty of the WISG.

Okay, point added.

L360, 364: Are these equation references supposed to be to A#?

Fixed.

L417: Is this Grobner (2025) from the main text?

Dropped. This information only available on calibration sheets not available to public.

---

## Author Comment (AC2)

In "Optimizing the precision of infrared measurements using the Eppley Laboratory, Inc. model PIR pyrgeometer", Michalsky et al. assess the performance of four different equations that are used to convert raw voltage measurements of pyrgeometers into irradiances during transfer calibrations. The authors conclude that the method by Philipona et al. (1995) is the preferred option for this purpose. Additionally, some analysis on the conversion of thermistor-measured resistance to temperature and on the consistency among different calibration events are provided. The topic is well suitable for publication in ATM. However, I think, the study has more potential and should first receive a careful revision considering the comments below.

General comments

The introduction could be improved. First, a general motivation on why accurate thermal infrared radiation measurements are required is missing. Although this might be obvious, it might help putting the later results into context. Paragraph added to introduction. Second, please add a short explanation on the measurement principle of the pyrgeometers and why the instrument temperature needs to be corrected for. Mostly unchanged except for emphasizing the need for careful receiver temperature measurements. Thereby, especially readers less familiar with such instruments will better understand the components in Fig. 1 and Eq. (1) and the difference between $T\_B$ and $T\_R$ Added notes in caption of Fig. 1.

What were the meteorological conditions during the instrument calibration? Was the calibration performed according to WRC standards (calibration coefficient C from outdoor measurements in clear-sky night-time conditions and coefficients $k\_i$ from lab experiments)? L 166-167 only mentions the exclusion of outliers and periods of precipitation. Please provide more information on the measurement site and the conditions during calibration (temperature, humidity, clouds). These are discussed in sections 3 and 5. Perhaps, meteorological data could complement the irradiance time series shown in Fig. 3. I think, the study can largely benefit from such data. Regarding Fig. 7, this data could be used to more accurately filter for specific conditions, such as cloudiness, temperature or humidity regimes, and assess, how different conditions affect the transfer calibration of the pyrgeometers. I would look forward to an extended analysis on this problem in Sect. 4 as a second focus of the study. See section 5.

I would suggest testing the consistency of the standard PIRs' calibration (Fig. 6) prior to calibration transfer to get an estimate of the impact of potential instrument instabilities on the transfer calibration. Fig. 6 seems to reveal that PIR 32909 is less stable than the other ones, especially regarding the 2024 calibration. This probably causes the underestimation in measured irradiance that is visible in Fig. 3 also for Philipona's coefficients. However, due to the significantly stronger underestimation for Albrecht's coefficients, I assume that Fig. 6 would also show larger differences for Albrecht. Consider showing Fig. 6 for both Albrecht and Philipona in conjunction with Fig. 3 (see also more detailed comment on Fig. 3 below). The results might

imply a possible exaggeration of instrument instabilities by Albrecht, which would further substantiate the preference for Philipona's method.

I made plots like Fig. 6 for the Albrecht and Cox WRC calibrations; they are similar to boxplots of Fig. 6 using Philipona coefficients with somewhat less scatter. Not sure how to explain this.

Based on the general comments above, consider modifying the general structure of the manuscript: My suggestion would be the following:

Motivation: Different equations for conversion of raw signal (voltages) to irradiance, all depending on body and dome temperature (measured by thermistor)

Sect. 2: evaluate different methods to convert thermistor resistance to temperature → no significant impact on irradiance

Sect. 3: evaluate equations to convert voltages to irradiances

3 regularly calibrated "standard" PIRs: first assess instrument stability and consistency of calibration events (Fig. 6) → PIRs generally stable, but 32909 least stable, impact of instability larger for Albrecht → preference Philipona (see general comment 3)

Transfer calibration standard PIRs → test PIRs (Figs. 4, 5) → preference Philipona

Modified Sect. 4 "Impact of meteorological conditions on precision of transfer calibration" (see general comment 2)

Furthermore, I would like to encourage the authors to carefully revise the text. The wording frequently sounds too colloquial, is imprecise, or lacks clarity. Sometimes, the language could be more concise. Some comments are given below, but there is room for more improvement.

General structure was not modified. Other comments addressed elsewhere.

Specific comments

Title: Consider changing to something like "Optimizing the precision of infrared radiation measurements by Eppley PIR pyrgeometers"

I'll keep what I have.

L 16-17: Is the cosine response important for the study? Rather mention more important instrument characteristics, such as the thermistors that are used for temperature measurements and referred to in L 20.

The cosine response point emphasizes that the PIR is intended to make hemispherical radiation measurement ala pyranometers.

L 18: This sentence lacks clarity. Do "standards" and "field units" refer to the standard pyrgeometers calibrated at WRC and the physical unit of irradiance, respectively? Please clarify.

Change made to as requested.

L 87-88: Please justify why k_0 is dropped.

There is no physical reason to have this term. Text added to make this point. BTW, the most extensive use of the Reda et al. equation is in the ARM program, which does not include it.

L 91-92: Avoid the term Steinhart-Hart equation here since it was not defined yet. Better "various versions of the Steinhart-Hart equation that have been used to convert …" → "various methods used to convert …".

Changed.

L 93: "three test PIRs" – later, the text mentions 6 instruments. Please clarify.

Changed.

Eq. (7): I think, the notation of the exponentiated logarithms is improper. I suggest using either $(\ln R)^2$ or $\ln^2(R)$. Furthermore, I would suggest including the quadratic term with coefficient c in Eq. (7) and set to 0 when necessary.

$(\ln R)^2$ used now, and $c$ coefficient changed to respond to another referee.

L 117: Is my understanding right that the regression (Eq. 7) to derive the coefficients a, b, (c), d is based on tabulated data of resistance and temperature provided by the manufacturer? At least, this statement is made in the conclusions section. If so, what is the benefit of performing the fit instead of simply interpolating between the tabulated values? Please add important information and be more accurate in describing what was done here and why.

The Steinhart-Hart equation is typically used to interpolate between the manufacturer's tabulated values. This section simply looks at how well other forms or sets of coefficients for this equation perform in fitting to these tabulated values. I suspect that the fits discussed in Fig.2 may indicate slight inaccuracies in the manufacturer's data. Many changes based on the other referee's comments hopefully have made this discussion clearer.

L 118-123: Consider altering the argumentation here and sticking to ohms when including the quadratic term into the regression. Although kiloohms could be skipped completely from this analysis, it might be worth mentioning the difference when the quadratic term is omitted in the following sentence. I like the idea of drawing attention to this discrepancy arising from the choice of the unit here and explain it mathematically. However, since this explanation is not the main scope of the study, I agree that the appendix is the right place to do that.

This was clarified (I think) by changing text based on the other referee's comments.

L 125: Is the reference Gröber (2025) accessible to the public? It would be good to know which coefficients are used for the corresponding fit. Maybe, a table listing the coefficients for all fits considered in Fig. 2 can be added. This table added.

| | a = | b = | c = | d = |
|---|---|---|---|---|
| Red (ohms) | 0.001029607 | 0.0002390769 | 0.0 | 1.567609E-07 |
| White (ohms) | 0.001020630 | 0.0002416721 | -2.47485E-07 | 1.64547E-07 |
| Blue (kiloohms) | 0.002732470 | 0.0002618082 | 3.162474E-06 | 1.645474E-07 |
| Gray (ohms) | 0.0010295 Better 0.0010293 | 0.0002391 | 0.0 | 1.568E-07 |
| Magenta ((ohms) | 0.00102972 | 0.00023906 | 0.0 | 1.5677E-07 |

L 126: "… but does well over the entire range." Please specify the temperature range referred to at the beginning of the section. Furthermore, "entire range" is imprecise here and should be changed to something like "entire temperature range analyzed here".

Fixed.

L 129-130: Could you justify your choice quantitatively? What does the temperature deviation mean in terms of irradiance?

Text added to summarize effect of any of these temperature estimates.

L 148: Was the calibration of the field/test PIRs only done for period shown in Fig. 3? According to the text, Fig. 3 is just an example. Please be more accurate in describing the calibration procedure and mention, which instrument was calibrated when.

Text added to make this clear.

Fig. 3: To put the irradiance measurements into context with the concurrent meteorological conditions, my suggestion is to add time series of relevant meteorological quantities to the time series of the irradiances, if available. Furthermore, only show the mean of the three instruments in the time series. The difference between the individual instruments (e.g. with respect to the mean) could be plotted in an additional histogram (or boxplots), from which the magnitude of potential over- or underestimation can be better quantified.

Histogram added to clearly make the points in the reviewer's comments.

L 163: How was "the mean IR irradiance of the three standards" obtained? How were the three WRC calibration (2018, 2022, 2024) combined to calculate the standards' irradiance? Please provide more information.

The first question was dealt with in response to the first reviewer. Text was added to clarify. If the second comment refers to Fig. 6 then same comment applies. We did not combine all three WRC calibration results to get a combined standard if that is what is meant.

L 188-189: I think, it's fine to only plot the results of two instruments. However, maybe summarize statistics for all instruments in a table and mention some key numbers in the text to corroborate the conclusions.

If one assumes that there are no significant differences in the calculation of infrared irradiances using the Philipona et al. (1995) formula versus each of the other three methods discussed in this paper, this assumption is rejected with 95% confidence in 15 of the 18 cases studied (six calibrated PIRs and three formulae). The three cases where the null hypothesis cannot be rejected with 95% confidence are for three of the six PIRs using the Reda et al. (2002) formula. Changes have been made to the text.

L 218-219: Is it fair to argue that the results are better or worse in Fig. 5 compared to Fig. 4? I mean, you are comparing to different references (standards calculated with Albrecht vs. Philipona) but none of these references really represents the truth. I think the comparison only shows that calibrating the test PIRs with Albrecht's equation can fit better to either the Albrecht- or the Philipona-calibrated standard and that the consistent application of Albrecht's method does not assure a minimized spread.

I just removed this sentence.

Fig. 7: How consistent are the results if you randomly split the data into equally sized subsets for calibration and validation?

We divided all six data sets and found results consistent with Fig. 7 except in one case where results were more consistent with using the entire data set when using either half of data to calibrate and test. Note that the 2024 calibration run was generally noisier than the 2023 run.

Appendix: Equation numbers 8–14 in the text → A1–A7

Fixed.

Text improvements

throughout: the use of written-out "infrared" and abbreviated "IR" is inconsistent. I suggest using the acronym "TIR" for thermal infrared, since the infrared also covers parts of the solar spectrum where the pyrgeometer is not sensitive.

Change made.

throughout: "degrees K" → "Kelvin"

Change made.

throughout: "standards" → "standard PIRs"

*Change made.*

    L 14: "The Eppley Model PIR is widely used …" → "The Eppley Precision Infrared Radiometer (PIR) is a widely used pyrgeometer …"

*Change made.*

    L 17: "equations in the literature" → "equations suggested by the literature"

*Change made.*

    L19-20: "… used to convert the resistance of the YSI 44021 thermistors used in PIRs for temperature measurements …" → "… used to convert the resistance measurements of the thermistors to temperatures …"

*Change made.*

    L 20, 31: skip "aka case"

*Change made.*

    L 29: "The Eppley model PIR pyrgeometer was developed …" → "The Eppley Precision Infrared Radiometer (PIR) is a pyrgeometer developed …"

*Change made.*

    L 29: skip "longwave"

*Change made.*

    L 30: "sky" → "atmosphere"

*Change made.*

    L 46: skip "L is the external incoming infrared irradiance" since L was defined above

*Change made.*

    Caption Fig. 1: "rays" → "radiation components on the thermopile surface" and "incoming infrared" → "incident atmospheric TIR irradiance".

*Change made.*

    L 112: "… where T is in degrees K and R is in ohms or kiloohms" → "… where T is the temperature in Kelvin and R is the measured resistance in Ohms or Kiloohms."

*Change made.*

L 118: "The least-squares fit to Eq. (7) … is the red line …" → "The least-squares fit to Eq. (7) … is indicated by the red line …"

Change made.

L 118-119 and 129-130: "full cubic" → "full cubic relationship"

Change made.

L 139: "we apply Eq. (2), (3), (4), and (6) to examine how well each performs …" → "we examine the performance of Eqs. (2), (3), (4), and (6) …"

Change made.

L 141: "Our three PIRs …" → "The three standard PIRs …"

Change made.

L 141-142 and throughout: be consistent with terms "PMOD" and "WRC".

Change made.

L 229-235: Most information in this paragraph is redundant, because it is already known. Suggestion to simplify: "The regular calibration of the standard PIRs at the WRC leads to different calibration results. Here, the consistency and repeatability of those calibration events is assessed." Also consider changing the structure (see general comments).

Change made.

L 306-307: "The three standard PIRs … are sent biennially to be calibrated …" → "The three standard PIRs … are biennially calibrated …"

Change made.

L 328: "In this paper, the World Infrared Standard Group (WISG) …" → "The WISG …"

Change made.

L 345-348: "The source of the difference …" → "This difference is due to numerical reasons, which are explained in the following."

Change made.